# Mutations in the *FOXO3* Gene and Their Effects on Meat Traits in Gannan Yaks

**DOI:** 10.3390/ijms25041948

**Published:** 2024-02-06

**Authors:** Youpeng Qi, Xiangyan Wang, Chune Zhu, Baohong Mi, Changze Cui, Shaopeng Chen, Zhidong Zhao, Fangfang Zhao, Xiu Liu, Jiqing Wang, Bingang Shi, Jiang Hu

**Affiliations:** Gansu Key Laboratory of Herbivorous Animal Biotechnology, College of Animal Science and Technology, Gansu Agricultural University, Lanzhou 730070, China; qiyp_gsau@163.com (Y.Q.); wxy9242022@163.com (X.W.); 18394174334@163.com (C.Z.); mi_baohong@163.com (B.M.); cuichang-ze0120@163.com (C.C.); 15095373670@163.com (S.C.); zhaozd@gsau.edu.cn (Z.Z.); zhaofangfang@gsau.edu.cn (F.Z.); liuxiu@gsau.edu.cn (X.L.); wangjq@gsau.edu.cn (J.W.)

**Keywords:** yak, *FOXO3*, SNP, meat quality

## Abstract

The *FOXO3* gene, a prominent member of the FOXO family, has been identified as a potential quantitative trait locus for muscle atrophy and lipid metabolism in livestock. It is also considered a promising candidate gene for meat quality traits such as Warner–Bratzler shear force (WBSF) and water holding capacity (WHC). The aim of this study was to identify sequence mutations in the *FOXO3* gene of yaks and to analyze the association of genotypes and haplotypes with meat traits such as WBSF and WHC. Quantitative reverse-transcriptase PCR (RT-qPCR) was applied to determine the expression levels of *FOXO3* in yak tissues, with the results revealing a high expression in the yak longissimus dorsi muscle. Exons of the *FOXO3* gene were then sequenced in 572 yaks using hybrid pool sequencing. Five single nucleotide polymorphisms were identified. Additionally, four effective haplotypes and four combined haplotypes were constructed. Two mutations of the *FOXO3* gene, namely C>G at exon g.636 and A>G at exon g.1296, were associated with cooked meat percentage (CMP) (*p* < 0.05) and WBSF (*p* < 0.05), respectively. Furthermore, the WBSF of the H2H3 haplotype combination was significantly lower than that of other combinations (*p* < 0.05). The findings of this study suggest that genetic variations in *FOXO3* could be a promising biomarker for improving yak meat traits.

## 1. Introduction

Yaks (*Bos grunniens*), native to the Tibetan Plateau and its surrounding areas, are endemic to those regions where they act as a source of meat, milk and fur for local herdsmen. China currently boasts a yak population of over 14 million, which accounts for more than 95% of the global total [1]. Since these animals live under peculiar conditions, characterized by high altitudes, extreme cold, lack of oxygen, high radiation and a short vegetative growing season, they produce a distinct type of meat that is known for its excellent quality. In fact, its protein, fat and amino acid content is even superior to that of beef [2]. However, the meat’s thicker muscle fibers, coarser texture and low intramuscular fat (IMF) content tend to lower its quality, which is a major determinant for the development of yak meat products [3]. Animal muscle development results from a balanced interplay of catabolism and synthesis (especially that of myosin and other proteins), which are influenced by a number of factors, such as environment, physiology, nutrition and genetics, with the latter exerting the greatest influence [4]. Over the past half a century, traditional animal breeding has applied quantitative genetics, line selection crossbreeding and other conventional techniques to continuously select and improve superior breeds, but it is now increasingly challenging to generate breakthrough hybrids with desirable traits. However, molecular marker-assisted breeding technology has revolutionized that process as, in addition to overcoming the limitations of traditional methods, it also helped to achieve greater precision in animal breeding. As a result, molecular marker-assisted selection of single nucleotide polymorphism (SNP) now represents a potent tool for enhancing the traits of superior livestock breeds [5]. Indeed, by taking advantage of genotype–phenotype correlations in organisms, specific phenotypes are selected to genetically improve livestock, with the process largely facilitated by the discovery of large numbers of SNPs markers. For instance, in studying the yak HSL gene, Wang et al. detected five SNP loci on exons 2 and 8 which were correlated with the Warner-Bratzler shear force (WBSF) [6]. Similarly, Jiang hu identified 12 and 4 SNPs that could be associated with body weight in male and female yaks, respectively, with 9 and 2 of these respective SNPs exerting a significant influence [7]. The above examples highlight the feasibility of applying SNPs to identify genetic loci that can subsequently facilitate the improvement of meat quality traits in yaks.

The FOXO class of transcription factors is involved in many vital activities of the body. For instance, overexpression of constitutively active *FOXO1* inhibits the differentiation of adult myoblasts into myotubes and, conversely, its inactivation partially restores C2C12 cells to a state where differentiation is inhibited [8]. FOXO is also known to be expressed in skeletal muscles where it regulates metabolism and may even promote muscle atrophy. In this context, a study involving transgenic mice showed that *FOXO1* overexpression in skeletal muscles caused severe myasthenia, with the in vitro and in vivo overexpression of *FOXO3* increasing the activity of the Atrogin-1 promoter to further accelerate myasthenia [9]. In addition FOXO is involved in the metabolic regulation of adipose tissue where it inhibits the differentiation of preadipocytes [10]. Existing research on the *FOXO3* gene is largely focused on human, mouse, horse, chicken and cow, just to name a few. For example, KiM J-R detected 7 polymorphic sites in the *FOXO3* gene of 24 Koreans [11], while Wang Ling’s study found that 8 polymorphic loci of the *FOXO3* gene were present in each of the cattle within a population. Similarly, through an RNA-Seq-based study of the high and low tail pectoral muscle of recessive white rock chickens (WRRh, WRRI) and Xinghua chickens (XHh, XHI), Chen Biao et al. found 18 SNPs that were associated with *FOXO3* [12]. In the mammalian ovary, the *FOXO3* gene can further be associated with follicular atresia by promoting apoptosis in ovarian granulosa cells [13]. The above findings suggest that this gene could be linked to significant polymorphism. Suppressing the expression of FOXO has been shown to inhibit protein hydrolysis and subsequently affect animal growth as well as meat quality traits [14]. In this context, Sun et al. who studied the relationship between *FOXO1* mutations and growth in Qinchuan cattle, found that five SNPs were associated with some growth traits in this particular cattle population [15].

Despite the above findings, limited attention has been given to the role of this gene in yaks as well as the impact of its genetic variation on meat traits. Given the key role of *FOXO3* in the process of muscle atrophy, it is hypothesized that this gene may also exert some effects on fleshy traits such as WBSF [16]. Therefore, this study aimed to analyze the genetic influence of the *FOXO3* gene on yak meat traits. In addition, using this gene as a focal point, genetic screening was performed on randomly selected yak populations in view of exploring how mutations were associated with meat traits.

## 2. Results

### 2.1. Genetic Characteristics of the FOXO3 Gene SNPs in Gannan Yaks

In the genomic sequencing of the *FOXO3* gene in Gannan yaks, the following five SNPs were identified as shown in Figure 1: g.636C>G (SNP1), g.660T>C (SNP2), g.1296A>G (SNP3), g.1413C>G (SNP4) and g.1699C>G (SNP5). The SNPs were genotyped using KASP; all SNPs had three genotypes (Figure 2).

The genotype frequencies, allele frequencies and genetic polymorphism for variations in the *FOXO3* gene of Gannan yaks are shown in Table 1. The most frequent genotypes for SNP1, SNP2, SNP3, SNP4 and SNP5 were CC, TC, AG, CG and CC, respectively, with all five SNPs of the Gannan population being in Hardy–Weinberg equilibrium (HWE) (*p* > 0.05). Based on polymorphic information content (PIC), SNP1, SNP2, SNP3 and SNP4 were found to exhibit moderate polymorphism (0.25 < PIC < 0.5), unlike SNP5, which showed low polymorphism (PIC < 0.25).

### 2.2. Gene Linkage Disequilibrium Analysis and Haplotype Construction in Gannan Yak’s FOXO3

Results of the linkage disequilibrium analysis of the five SNPs in the *FOXO3* gene of yaks are shown in Figure 3. Linkage disequilibrium (LD) refers to the non-random association of alleles at different loci, and in this study, LD was determined using r^2^, which showed strong associations among the SNPs (r^2^ was > 0.33), except for SNP1 and SNP5.

Four haplotypes (H1–H4), with frequencies exceeding 3%, were identified as shown in Table 2. Haplotype H1 had the highest frequency of 44.2%, followed by H2 with a frequency of 42.2%. H3 and H4 had the lowest frequencies of 4.8% and 4.7%, respectively. These four haplotypes resulted in various haplotype combinations, with 5% of all individuals having the following four combinations: H1H1, H1H2, H2H2 and H2H3.

### 2.3. Association of FOXO3 Genotypes with Meat Traits in Gannan Yaks

Table 3 shows the results of the association analysis between *FOXO3* genotypes and meat quality traits, such as WBSF, CMP and WHC. In Gannan yaks (the number of individuals with SNP1 locus GG and SNP5 locus GG was low and, hence, no ANOVA was performed). Different genotypes were found to affect WBSF and WHC. For instance, the AG type of SNP3 exhibited a significantly lower shear force than the other genotypes (*p* < 0.05), while the CC genotype of SNP1 and SNP5 was associated with a significantly higher WHC (*p* < 0.05). Thus, the results suggest that in Gannan yaks, the AG genotype of the SNP3 locus as well as the GC genotype of the SNP1 and SNP5 loci of the *FOXO3* gene could be used as genetic markers for WBSF and WHC, respectively.

### 2.4. Effects of Haplotype Combinations on the Meat Traits of Gannan Yak Meat

Association analysis was also performed for different haplotype combinations of the *FOXO3* gene (more than 2% of the sampled population) meat traits, with the results presented in Table 4. Overall, haplotype combinations H2H2 and H2H3 showed a significantly lower shear force than H1H1 and H1H2 (*p* < 0.05). Combined analysis subsequently showed that individuals with the H1H2 haplotype combination were dominant and, hence, this combination could be used as a molecular marker for meat traits in Gannan yaks.

Table 5 further explores how the presence/absence of specific haplotypes was linked to meat traits. In this case, the presence of the H2 haplotype was associated with decreased WBSF (*p* = 0.009), with this correlation remaining significant when other haplotypes (where *p* < 0.2) were included into the models. However, REA, CMP or WHC were not associated with any specific haplotypes in Gannan yaks.

### 2.5. Expression Analysis

The relative expression of *FOXO3* gene mRNA was determined for six different yak tissues, with the results, shown in Figure 4, indicating that the gene was commonly expressed across all the tissues. In particular, the highest expression was observed in the longest dorsal muscle, followed by the large intestine, adipose, lung, heart and small intestine.

## 3. Discussion

This study investigated the relationship between *FOXO3* gene sequence mutations and meat traits in Gannan yaks. The results confirmed the presence of genetic variations in this gene and highlighted their significant effects on the WBSF of the yaks, thereby suggesting the significance of undertaking further studies on *FOXO3* gene mutations in various yak breeds.

In this study, five variation loci were identified in the exonic region of the *FOXO3* gene of Gannan yaks. Tsai-Chung Li et al., who studied the relationship between *FOXO3* mutations as well as the effect of physical activity on physical performance, found two SNPs in the *FOXO3* gene [17]. Similarly, Soerensen identified 15 SNPs in the *FOXO3* gene when studying the association of the gene’s SNPs with aging in Danish individuals of advanced age [18]. Overall, the presence of these SNPs, which reflect a high degree of genetic variation in a population, is indicative of genetic richness. Polymorphic information content (PIC) is dependent on the number of alleles tested and their frequency of distribution. In this study, except for SNP5, the remaining SNP loci detected in the *FOXO3* gene of Gannan yaks exhibited moderate polymorphism, thereby indicating a high genetic richness that can serve as a solid foundation for genetic variation. Furthermore, all loci were found to be in Hardy–Weinberg equilibrium, and this reflected the historical breeding practices of Gannan yaks, involving minimal selection and improvement, in alpine grazing areas.

Exon sequences are relatively conserved, with the probability of mutations being only one-fifth that of introns. It has been found that conserved sequence fragments often correspond to functionally important regions, and therefore, nsSNPs located at highly conserved sites are more likely to become functional SNPs than those found at non-conserved sites [19]. A genome-wide association analysis of 193 Nellore cattle revealed that *FOXO3*’s regulation of processes related to lipid metabolism may involve major adipogenic genes, such as PPARy and C/EBPa [20]. Similarly, it has been shown that polymorphism in the second exon of the *FOXO3* gene was significantly associated with intramuscular fat content, shear force, carcass slant length, back fat thickness between ribs 6 and 7, marbling and water loss rate in pigs [21]. In addition, Wang qi identified three SNPs (97538(G/A), 98109(A/G), and 98226(G/C)) that were significantly different from the production traits, in the exons of *FOXO3* [22]. In this study, five SNPs were identified in the *FOXO3* gene of yaks, some of these SNPs were similar to the above studies. Two of these variation loci, namely the CG genotype of SNP1 and the AG genotype of SNP3, were associated with significantly lower WHC and WBSF in Gannan yaks, respectively (*p* < 0.05). This was in accordance with the findings of other studies and supported the hypothesis that mutations could influence the translation and expression of the *FOXO3* gene, thereby affecting the quality of yak meat. Meat traits are complex quantitative traits that are controlled by multiple genes. While individual variation loci play a role, the combined effects of multiple loci have a greater impact on phenotypic trait mutation [23].

Analysis of haplotype combinations takes into account non-allelic interactions and linkage disequilibrium between multiple mutation loci, and this approach offers greater statistical power than statistical analysis between alleles [24]. Recently, haplotype analysis has gained prominence in the study of complex genetic phenotypes, with a haplotype defined as a set of closely linked SNPs that is inherited as a unit on a single chromosome. Understanding the haplotypes of multiple SNPs within a gene can provide more information on genotype–phenotype associations than analyzing individual SNPs alone. In a study of the *ANK1* gene in Gannan yaks, Hu et al. found that the presence/absence of specific haplotypes and haplotype copy numbers influenced carcass weight, muscle water loss and shear force [25]. In addition, Wang Ling found that different haplotype combinations of the *FOXO1* gene significantly affected the shear force, muscle fiber diameter and muscle fiber density of common cattle. In the present study, the combination of H2H3 haplotypes exhibited better WBSF, hence, suggesting that this haplotype could be a potential molecular marker for future efforts to improve yak meat quality. In summary, the *FOXO3* gene exhibited promising effects on meat traits in yaks.

Previous research revealed widespread expression of the *FOXO3* gene in chickens and yaks, especially in muscle tissues, heart and testes [12,22]. The current findings were aligned with these studies, noting the highest expression of the *FOXO3* gene in the longissimus dorsi of yaks. We hypothesize that the *FOXO3* gene may be involved in muscle formation and developmental processes. Eelen et al. showed that the *FOXO3* gene is mainly expressed in skeletal muscle and is associated with muscle atrophy [26]. Muscle atrophy is one of the symptoms of a variety of pathological conditions. Fasting, disease, or denervation of specific muscle groups will lead to systemic muscle atrophy. It is mainly caused by the accelerated breakdown of muscle proteins and is related to the activation of the ubiquitin/proteasome pathway [27]. The reduction of the IGF-1/PI3K/AKT signaling pathway will increase muscle atrophy, leading to the induction of two genes, MAFbX (muscle atrophy F box, Atrogin-1) and MuRF-1 (muscle RING finger 1) [28]. Furuyama and Imae showed that the expression levels of *FOXO1* and *FOXO3* would increase in a state of muscle starvation and glucocorticoid treatment, promoting muscle atrophy [29,30]. When cultured myotubular cells are subjected to starvation and glucocorticoids, the IGF-1-PI3K-AKT pathway is reduced, FOXO expression levels are increased, as are Atrogin-1 and MuRF-1 expression levels, and muscle atrophy ensues [31]. In a healthy model of muscle atrophy, nuclear localization of *FOXO3* is increased, which is required for the Atrogin-1 promoter [32]. Overexpression of *FOXO3* in vitro and in vivo will increase Atrogin-1 promoter activity, and transgenic mice overexpressing *FOXO1* in skeletal muscle result in severe muscle atrophy [9]. Leger found from muscles isolated from patients with amyotrophic lateral sclerosis that, despite increased levels of Atrogin-1 and reduced AKT phosphorylation, *FOXO1* and *FOXO3* expression and nuclear localization were unchanged, whereas in healthy people altering muscle mass affected *FOXO1* expression and nuclear localization. Exercise induced muscle hypertrophy, leading to increased AKT phosphorylation and reduced *FOXO1* in the nucleus; conversely, no exercise induced muscle atrophy, leading to reduced AKT phosphorylation and increased *FOXO1* in the nucleus [33]. The above findings suggest that the *FOXO3* gene affects muscle development and mediates muscle atrophy.

This study identified four valid haplotypes and four haplotype combinations. In particular, individuals with the H2H3 combination showed better tenderness, thereby suggesting that this haplotype could be a potential molecular marker for future endeavors to improve the meat quality of yaks. This study is the first to determine the relationship between *FOXO3* sequence variation and yak meat quality traits, which is of great significance for yak meat quality improvement. The next work should be further validated with *FOXO3* gene function. Overall, the *FOXO3* gene showed promising effects on meat quality traits in yaks.

## 4. Materials and Methods

### 4.1. Animal Selection

All animal experiments were conducted in accordance with the guidelines for the care and use of experimental animals established by the Ministry of Science and Technology of the People’s Republic of China (Approval number 2006-398) and was approved by the Animal Care Committee of Gansu Agricultural University, Lanzhou, China. A total of 572 Gannan yaks (6 years of age, *n* = 523; 5 years of age, *n* = 49) were randomly selected from the Gannan Tibetan Autonomous Prefecture, Gansu Province, China. All of these yaks grazed on the plateau pasture at an altitude of 2500–4000 m all year round, freely feeding on grass and drinking water. Prior to slaughter, 25 mL of jugular vein blood was collected from each yak using a sodium heparin tube. These samples were kept at −80 °C until required for blood genomic DNA extraction.

Six tissues, including heart, lungs, small intestine, longissimus dorsi, large intestine and subcutaneous fat were carefully collected from three six-year-old male Gannan yaks within 30 min of slaughter in October, which is considered to be the beginning of the cold season of the Qinghai–Tibetan Plateau. All samples were immediately snap-frozen in liquid nitrogen and then stored at −70 °C.

At 48 h postmortem, the rib eye area (REA; cm^2^) was determined from the longissimus dorsi between the 12th and 13th ribs using sulfate paper and estimated with a planimeter. Samples of the longissimus dorsi muscle, between the 11th and 12th thoracic rib of the right carcass side, were then collected and quickly frozen for storage at −18 °C until required for measuring the Warner–Bratzler shear force (WBSF; kg), water holding capacity (WHC; %) and cooked meat percentage (CMP; %). WHC was measured using a modified filter paper press method described by Liu et al. [34], while WBSF and CMP were determined according to Shackelford et al. [35] and Honikel [36], respectively.

### 4.2. DNA Extraction and PCR

After thawing the yak blood samples, DNA was extracted according to the instructions of the Tiangen Blood Genomic DNA Extraction Kit (Centrifugal Column Type, TIANGEN BIOTECH, Beijing, China). DNA concentration, integrity and purity were then tested using 1.5% agarose gel electrophoresis and an ultra-microvolume (NanoDrop^TM^ spectrophotometer (one/one^c^ Thermo Scientific, Waltham, MA, USA), respectively, with only suitable samples selected for detecting mutations.

The yak *FOXO3* gene sequence (ENSBMUG00000019087) was queried in the Ensembl database and primers (Table 1), designed using Primer 5.0 software, were then synthesized by Qing ke Biological Co., Ltd. (Xi’an, China). Thirty unrelated samples were randomly selected from the test yak genomic DNA to amplify the exonic region of the *FOXO3* gene as well as the 5′UTR, exons 1–2 and 3′UTR regions. PCR, performed in 20-µL reaction volumes, included 0.8 µL of DNA template (100 ng/µL), 0.8 µL of upstream and downstream primers (10 µmol) each, 10 µL of Taq DNA polymerase and 7.6 µL of ddH2O. The amplification procedure was then performed as follows: pre-denaturation for 5 min at 94 °C, followed by 35 cycles, each with a 30 s denaturation at 94 °C, a 30 s annealing at 60 °C and an extension for 30 s at 72 °C. The reaction ended with a final extension of 10 min at 72 °C. The amplification products of the 30 samples were taken (2 µL), and the amplified regions of each primer pair were built into separate mixing pools and sent to Shanghai Bioengineering Co., Ltd. (Shanghai, China).

### 4.3. Genotype Testing

After determining the mutation sites by mixed-pool sequencing, genomic samples of yaks were sent to Jing Tai Biological Co., Ltd. (Wuhan, China) for genotyping by the KASP method. The SNP sites were detected fluorescently in three PCR reactions using a primer mix containing two forward with two universal tags as well as a Master mix containing two detection primers with different fluorescent signals. The first PCR reaction involved denaturation of the DNA template, binding to matching KASP primers, annealing, extension and detection of primer sequences that were added. For the second PCR reaction, complementary strands of allele-specific terminal sequences were synthesized, while the third PCR reaction involved an exponential growth of the detection primer as the PCR reaction proceeds to generate and detect the fluorescent signal. At the end of the reaction, fluorescence data were read using a fluorescence resonance energy transfer (FRET) enzyme-labeled instrument before drawing a scatter plot in Microsoft Excel 2016 software.

### 4.4. RNA Extraction and RT-qPCR

Total RNA was extracted from yak tissues (heart, lungs, small intestine, Longissimus doris, large intestine and subcutaneous fat) using Trizol (Vazyme, Nanjing, China) reagent according to available instructions before assessing the concentration and integrity of the extracted RNA by spectrophotometry and agarose gel electrophoresis (1.5). This was followed by reverse transcription using gDNA reagent and the Prime Script^TM^ RT kit (Vazyme, Nanjing, China). RT-qPCR was then performed in 20 µL reaction volumes containing 0.8 µL of cDNA, 0.25 µM of each primer, 10.0 µL of AceQ qPCR SYBR^®^ Green Premix (Vazyme, Nanjing, China) and 0.4 µL of ROX Reference Dye 2, with the reaction steps being as follows: pre-denaturation at 95 °C for 30 s was followed by 40 cycles, each consisting of denaturation for 5 s at 95 °C and extension for 30 s at 60 °C. Melting curve analysis was then performed for 5 s at 95 °C and for 1 min at 60 °C. The reaction ended with cooling at 50 °C for 30 s. Gene expression was subsequently determined with the 2^−ΔΔCt^ method.

### 4.5. Statistical Analysis

Genotypic and allele frequency, homozygosity, heterozygosity, number of effective alleles, PIC as well as HWE of *FOXO3* SNPs in yaks were calculated in Microsoft Excel 2016 software. SNPs were analyzed for haplotype and linkage disequilibrium (LD) using SHEsis (http://shesisplus.bio-x.cn/SHEsis.html, accessed on 11 March 2023), with only haplotypes with a frequency exceeding 3% subsequently being retained. A general linear model (GLM) was also used in SPSS 26.0 to analyze the correlation of genotypes, haplotype combinations and haplotypes with meat traits as follows:Y_ijkl_ = μ + G_i_ + S_k_ + F_l_ + e_ijkl_, Y_ijkl_ = μ + D_i_ + S_k_ + F_l_ + e_ijkl_, Y_ijkl_ = μ + H_i_ + S_k_ + F_l_ + e_ijkl_

In the above formula, Y represents the trait phenotype value; μ is the overall mean; G_i_ is the genotype effect; D_i_ is the haplotype combination effect; H_i_ is the haplotype effect; S_k_ is the sex effect; F_l_ is the field effect; and e_ijkl_ is the random error.

## 5. Conclusions

Five SNPs were identified in the *FOXO3* gene of yaks. Subsequent association analysis of two specific polymorphisms (g.636C>G and g.1296A>G) revealed a significant influence on WHC and WBSF. In addition, significant effects of the H2H3 haplotype on the WBSF of yak meat were noted. This report provides evidence of the potential suitability of the *FOXO3* gene as a molecular marker for meat traits in view of improving meat traits of yaks through further marker-assisted selection.

## Figures and Tables

**Figure 1 ijms-25-01948-f001:**
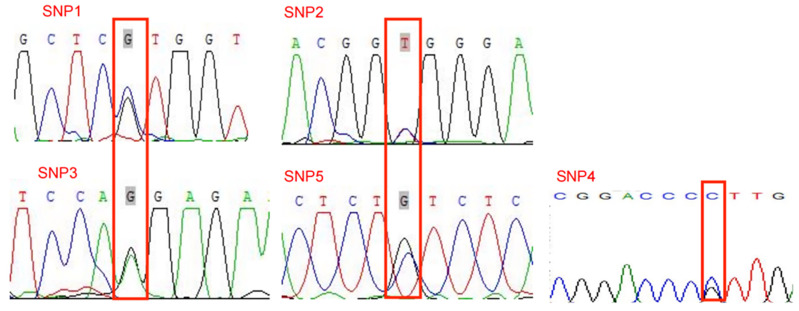
SNPs in *FOXO3* gene in yak.

**Figure 2 ijms-25-01948-f002:**
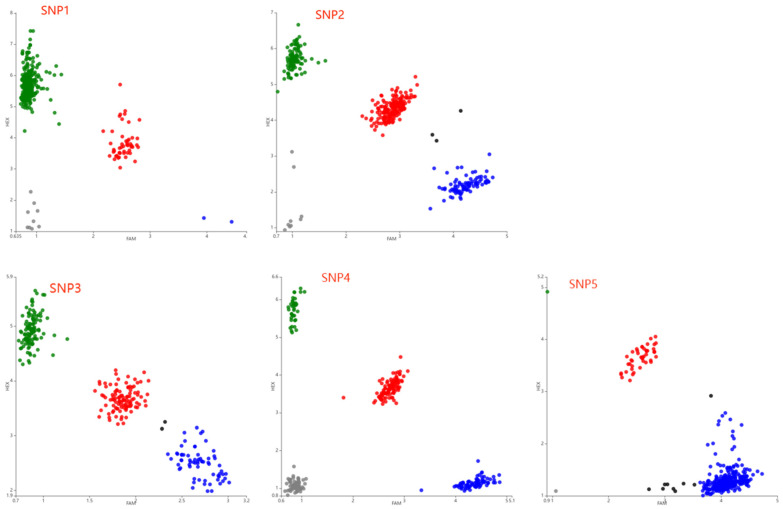
Results of Kompetitive allele-specific PCR (KASP) genotyping assay for the five positions in the *FOXO3* gene of Gannan yaks. The red, blue and green dots in each SNP site indicate different genotypes as follows: SNP1: GC, GG and CC genotypes, respectively; SNP2: CC, TC and TT genotypes, respectively, SNP3: AG, GG and AA genotypes, respectively; SNP4 and SNP5: CG, CC and GG genotypes, respectively. The black dots indicate negative controls and grey dots indicate unavailable measurements.

**Figure 3 ijms-25-01948-f003:**
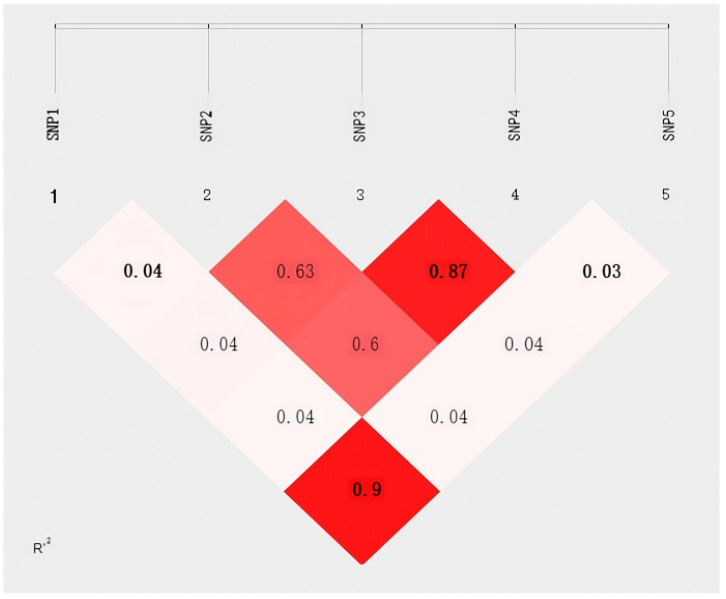
The linkage analysis disequilibrium of *FOXO3* gene in yaks. The higher two loci are in linkage disequilibrium, the darker the color will be.

**Figure 4 ijms-25-01948-f004:**
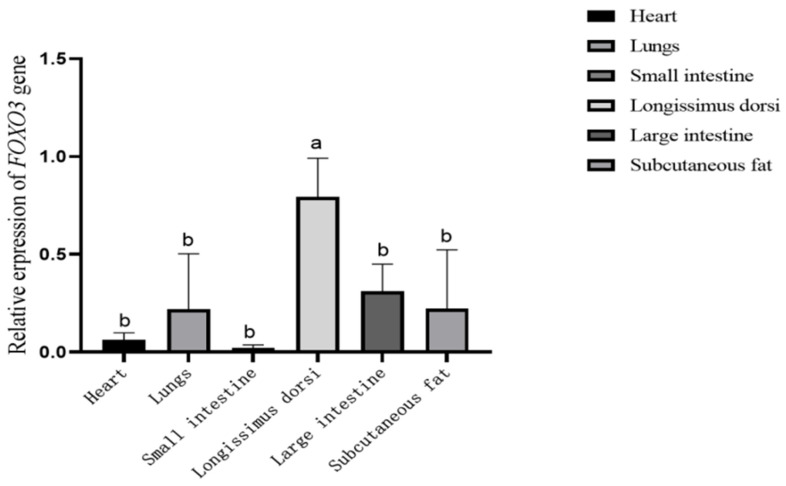
The relative expression of *FOXO3* in six yak tissues. The different lower-case letters above the columns indicate significant differences (*p* < 0.05).

**Table 1 ijms-25-01948-t001:** Genetic polymorphism analysis of *FOXO3* gene in yak.

SNP	Genotype Frequency			Allele Frequency		HWE	Genetic Polymorphism			
	Genotype	Number	Frequency	Allele	Frequency		PIC	He	Ne	HO
SNP1	CC	476	88.80	C	94.41	0.27	0.13	0.14	1.16	0.85
	GC	57	10.63	G	5.87					
	GG	3	0.559							
SNP2	CC	164	30.59	C	54.94	0.69	0.37	0.49	1.99	0.50
	TC	261	48.69	T	45.06					
	TT	111	20.70							
SNP3	AA	159	29.66	A	54.19	0.68	0.37	0.49	1.96	0.50
	AG	263	49.06	G	45.81					
	GG	114	21.26							
SNP4	CC	183	34.14	C	57.46	0.67	0.36	0.48	1.92	0.52
	CG	250	46.64	G	42.54					
	GG	103	19.21							
SNP5	CC	481	89.73	C	94.68	0.21	0.098	0.10	1.11	0.89
	CG	53	9.88	G	5.32					
	GG	2	0.37							

**Table 2 ijms-25-01948-t002:** Haplotype analysis of *FOXO3* gene mutation sites in yak.

Haplotype	Tag SNP	Frequency/%
H1	CTACC	44.2%
H2	CCGGC	42.2%
H3	GCACG	4.8%
H4	CCACC	4.7%

**Table 3 ijms-25-01948-t003:** Association analysis of *FOXO3* genotype meat traits in yaks.

SNP	Genotype	Meat Traits				
		Number	WBSF/kg	CMP/%	WHC/%	REA/cm^2^
SNP1	CC	476	5.51 ± 0.10	65.95 ± 0.40	21.31 ± 0.39 ^a^	31.31 ± 0.59
	CG	57	5.48 ± 0.20	66.07 ± 0.81	19.69 ± 0.78 ^b^	31.68 ± 1.19
	GG	3				
	*p*		0.845	0.382	**0.030**	0.739
SNP2	CC	164	5.47 ± 0.13	67.15 ± 0.53	21.58 ± 0.51	31.27 ± 0.78
	TC	261	5.44 ± 0.12	66.35 ± 0.48	20.82 ± 0.47	31.10 ± 0.71
	TT	111	5.69 ± 0.14	66.68 ± 0.59	21.31 ± 0.57	31.83 ± 0.86
	*p*		0.259	0.335	0.335	0.725
SNP3	AA	159	5.68 ± 0.13 ^a^	66.37 ± 0.54	21.04 ± 0.52	31.56 ± 0.80
	AG	263	5.35 ± 0.11 ^b^	66.50 ± 0.47	20.81 ± 0.45	31.53 ± 0.69
	GG	114	5.63 ± 0.14 ^a^	67.43 ± 0.59	22.08 ± 0.57	30.70 ± 0.87
	*p*		**0.031**	0.232	0.098	0.614
SNP4	CC	183	5.65 ± 0.1 3	66.33 ± 0.51	21.06 ± 0.50	31.57 ± 0.76
	CG	250	5.37 ± 0.12	66.54 ± 0.48	20.78 ± 0.46	31.45 ± 0.70
	GG	103	5.58 ± 0.15	67.50 ± 0.61	22.15 ± 0.59	30.76 ± 0.90
	*p*		0.084	0.205	0.086	0.697
SNP5	CC	481	5.52 ± 0.10	66.72 ± 0.40	21.30 ± 0.39	31.32 ± 0.59
	GC	53	5.42 ± 0.21	66.27 ± 0.84	19.78 ± 0.81	31.51 ± 1.23
	GG	2				
	*p*		0.605	0.569	**0.049**	0.876

Bold values indicate *p* < 0.05; data in the same column with different lowercase letters on the shoulders indicate significant differences (*p* < 0.05). *p* is derived from the General Linear Mixed Models (GLMMs). REA: rib eye area, WBSF: Warner–Bratzler shear force, WHC: water holding capacity, and CMP: cooked meat percentage.

**Table 4 ijms-25-01948-t004:** Association of *FOXO3* haplotypes with meat traits in yaks.

Diplotypes	Meat Traits				
	Number	WBSF/kg	CMP/%	WHC/%	REA/cm^2^
H1H1	101	5.55 ± 0.16 ^a^	67.53 ± 0.64	22.19 ± 0.62	30.78 ± 0.90
H1H2	203	5.46 ± 0.14 ^a^	66.41 ± 0.54	20.77 ± 0.52	30.92 ± 0.76
H2H2	119	5.10 ± 0.16 ^b^	66.85 ± 0.60	21.59 ± 0.59	31.78 ± 0.85
H2H3	32	4.89 ± 0.26 ^b^	66.56 ± 1.05	21.10 ± 1.03	30.46 ± 1.47
*p*		**0.014**	0.422	0.170	0.666

Bold values indicate (*p* < 0.05); data in the same column with different lowercase letters on the shoulders indicate significant differences (*p* < 0.05). *p* is derived from the General Linear Mixed Models (GLMMs). REA: rib eye area, WBSF: Warner–Bratzler shear force, WHC: water holding capacity and CMP: cooked meat percentage.

**Table 5 ijms-25-01948-t005:** Correlation of haplotype present/absent of *FOXO3* gene with meat traits in yaks.

Trait (Unit)^2^	Haplotype	*n*		Single-Haplotype Model		*p*	Multi-Haplotype Model			*p*
		Present	Absent	Present	Absent		Other Haplotypes in Model	Present	Absent	
WBSF/kg·f	H_1_	345	149	5.58 ± 0.13	5.53 ± 0.15	0.715	H2, H3	5.42 ± 0.15	5.50 ± 0.16	0.563
	H2	338	156	5.45 ± 0.12	5.84 ± 0.15	0.009	H3	5.32 ± 0.15	5.59 ± 0.16	0.041
	H3	47	447	5.19 ± 0.25	5.60 ± 0.12	0.079	H2	5.34 ± 0.22	5.56 ± 0.11	0.290
	H4	46	448	5.70 ± 0.25	5.56 ± 0.12	0.550	H2, H3	5.19 ± 0.25	5.46 ± 0.14	0.220
CMP/%	H1	345	149	66.49 ± 0.46	67.09 ± 0.56	0.270				
	H2	338	156	66.87 ± 0.46	66.27 ± 0.57	0.262				
	H3	47	447	65.66 ± 0.90	66.69 ± 0.43	0.974				
	H4	46	448	65.36 ± 0.90	66.80 ± 0.43	0.096				
WHC/%	H1	345	149	21.04 ± 0.78	20.19 ± 0.65	0.262				
	H2	338	156	20.86 ± 0.64	19.62 ± 0.79	0.100				
	H3	47	447	21.18 ± 1.25	20.42 ± 0.60	0.524				
	H4	46	448	18.45 ± 1.26	20.64 ± 0.61	0.070				
REA/cm^2^	H1	345	149	31.08 ± 0.84	30.82 ± 0.71	0.752				
	H2	338	156	30.84 ± 0.86	30.95 ± 0.70	0.889				
	H3	47	447	30.79 ± 0.65	32.44 ± 1.36	0.200				
	H4	46	448	31.62 ± 1.37	30.86 ± 0.66	0.558				

## Data Availability

The data presented in this study are available upon request from the corresponding author.

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
