# Peer review of "Mutations in the *FOXO3* Gene and Their Effects on Meat Traits in Gannan Yaks"

_ijms, 2024, doi:10.3390/ijms25041948_

Round 1
Reviewer 1 Report
Comments and Suggestions for Authors
The report by Youpeng Qi and colleagues describes the characterization of FoxO3 genotypes in 572 domestic Yaks (Bos grunniens) and the expression of the same gene in different tissues.
The analysis of five SNPs was performed, and their related haplotypes were associated with meat technological traits.
Although some results are interesting, the general structure of the work looks confusing, the link between the SNPs and the FOXO3 expression analysis is unclear, and the biological interpretation of the findings is incomplete.
Indeed, four of the five SNPs detected in exon II of FOXO3 (named 1 to 4 in the manuscript), are silent mutations that don’t result in any change in the protein structure. (By the way, the authors should recognize that SNPs 3 and 4 were already described by Wang, 2020, cited in references)
The only explanation of a possible influence of SNPs 1-4 on FOXO3 pathway would be a change in the expression level of the transcript, due to a hypothetic destabilization of the mRNA or a codon-usage influence.
To demonstrate this, the expression level of the gene should be analyzed in a target tissue (e.g. longissimus dorsi) of a cohort of individuals carrying the different genotypes.
Conversely, the detection of the (rare) SNP 5 is worth consideration. Indeed, it looks responsible for one of the few FOXO3 aminoacidic variation between Bos taurus and Bos grunniens (the rare SNP5 “T” genotype reverts the protein sequence to that of cattle).
This mutation by itself could be an interesting target for assisted selection, even though its rareness suggests a poor fitness for the harsh environment where yaks live.
Minor remarks
Fig.1 should be organized in a single panel of better quality and the headings of the five chromatograms should be corrected (at present they look wrong)
Comments on the Quality of English LanguageThe manuscript describes interesting results but the structure of the work is confused and the conclusion should be redrawn.
Author Response
The report by Youpeng Qi and colleagues describes the characterization of FoxO3 genotypes in 572 domestic Yaks (Bos grunniens) and the expression of the same gene in different tissues.
The analysis of five SNPs was performed, and their related haplotypes were associated with meat technological traits.
Although some results are interesting, the general structure of the work looks confusing, the link between the SNPs and the FOXO3 expression analysis is unclear, and the biological interpretation of the findings is incomplete.
Indeed, four of the five SNPs detected in exon II of FOXO3 (named 1 to 4 in the manuscript), are silent mutations that don’t result in any change in the protein structure. (By the way, the authors should recognize that SNPs 3 and 4 were already described by Wang, 2020, cited in references)
The only explanation of a possible influence of SNPs 1-4 on FOXO3 pathway would be a change in the expression level of the transcript, due to a hypothetic destabilization of the mRNA or a codon-usage influence.
To demonstrate this, the expression level of the gene should be analyzed in a target tissue (e.g. longissimus dorsi) of a cohort of individuals carrying the different genotypes.
Conversely, the detection of the (rare) SNP 5 is worth consideration. Indeed, it looks responsible for one of the few FOXO3 aminoacidic variation between Bos taurus and Bos grunniens (the rare SNP5 “T” genotype reverts the protein sequence to that of cattle).
This mutation by itself could be an interesting target for assisted selection, even though its rareness suggests a poor fitness for the harsh environment where yaks live.
Minor remarks
Thank you for your review and suggestions on our manuscript. Those comments are all valuable and very helpful for revising and improving our paper. We have studied comments carefully and have made correction which we hope meet with approval.
Muscle tenderness is generally reflected by WBSF; the lower WBSF, the more tender the muscle.On the one hand, correlation analysis revealed that SNPs (g.636C>G, g.1296A>G) in the FOXO3 gene were associated with yak muscle shear force and water loss, and it was hypothesised that mutations at the g.636C>G, g.1296A>G loci might affect yak muscle tenderness.On the other hand, high expression of a specific gene is a gene-specific enhancement response, implying that a particular gene has a higher concentration of activity than other genes during a particular activity.In this study, FOXO3 gene was highly expressed in the longest dorsal muscle tissue of yaks, and we hypothesised that FOXO3 gene was highly expressed through the muscle tissue to produce more muscle proteins, which made the activity of the gene more prominent, and thus affected the quality of yak meat.
Thank you for your suggestion to analyse gene expression in genotypically distinct groups, we will listen to your suggestion and this part will be the focus of our follow-up work.
Fig.1 should be organized in a single panel of better quality and the headings of the five chromatograms should be corrected (at present they look wrong)
Thanks for the suggestion, I'm catching up on optimising the charts and will upload the optimised charts to the editor later.
Reviewer 2 Report
Comments and Suggestions for Authors
I am convinced that the authors put a lot of effort into this study. Unfortunately, based on the study's current state, I would recommend to critically check its structure and proofread it on a linguistic level, and then resubmit it. I hope my input is of assistance to the authors.
In its current state, the manuscript is not at an acceptable standard of English. In some paragraphs, there are so many problems with the sentence structure that I was not able to understand the intentions of the authors. It is necessary to revise the manuscript (avoid tense switching; check appropriate punctuation, colour (black, red, blue?) conjugation, and typography).
Below are a few examples of issues in the study. This list is certainly not exhaustive.
line 11: „, therefore,” instead of „and”
line 11: “aim” instead of “aimed”
line 15-17: I recommend a more logical wording and connection of these two sentences.
line 21: insert space before bracket
line 21: Missing or unnecessary spaces: (P=0.030) vs. line 24 ( P = 0.014)
line 49-54: This section is professionally confusing; the genetic marker results in direct genetic selection. Incorrect wording: “molecular marker-assisted selection of SNPs” is
line 54: “SNPs” instead of “SNPS”
line 54: “Page et al. 2002 (year)” ??
line 55: “which were genotyped by the resource population” ??
line 56: “in both populations”. Which ones?
line 70: “HRIBAL” - it would be Hribal et al. (2003)?
Author Response
I am convinced that the authors put a lot of effort into this study. Unfortunately, based on the study's current state, I would recommend to critically check its structure and proofread it on a linguistic level, and then resubmit it. I hope my input is of assistance to the authors.
In its current state, the manuscript is not at an acceptable standard of English. In some paragraphs, there are so many problems with the sentence structure that I was not able to understand the intentions of the authors. It is necessary to revise the manuscript (avoid tense switching; check appropriate punctuation, colour (black, red, blue?) conjugation, and typography).
Below are a few examples of issues in the study. This list is certainly not exhaustive.
line 11: „, therefore,” instead of „and”
Thank you for your suggestion, I have revised it.
line 11: “aim” instead of “aimed”
Thank you for your suggestion, I have revised it.
line 15-17: I recommend a more logical wording and connection of these two sentences.
Thank you for your suggestion, I have reworked the sentence.
line 21: insert space before bracket
Thank you for your suggestion, I have reworked it.
line 21: Missing or unnecessary spaces: (P=0.030) vs. line 24 ( P = 0.014)
Thank you for your suggestion, I have reworked it.
line 49-54: This section is professionally confusing; the genetic marker results in direct genetic selection. Incorrect wording: “molecular marker-assisted selection of SNPs” is
Thank you for such professional guidance, I have reviewed the literature carefully and have revised it.
line 54: “SNPs” instead of “SNPS”
Thank you for your suggestion, I have revised it.
line 54: “Page et al. 2002 (year)” ??
Thank you for your suggestion, I have revised it.
line 55: “which were genotyped by the resource population” ??
Thank you for your suggestion, I have revised it.
line 56: “in both populations”. Which ones?
Thank you for your suggestion, I have revised it.
line 70: “HRIBAL” - it would be Hribal et al. (2003)?
Thank you for your suggestion, I have revised it.
Thank you for being able to revise my manuscript so carefully, a million thanks.
Reviewer 3 Report
Comments and Suggestions for Authors
The article titled “Variation in FOXO3 Gene and Its Effect on Meat Traits in Gannan Yaks” aims to associate some of the meat and carcass quality traits to specific SNPs and haplotypes in Gannan Yaks. The study used a large number of individuals along with phenotypic data, which is a very big advantage for this type of study. However, there are a number of issues that need to be addressed and clarified, especially in relation to the material and methods used. The results and discussion also need to be reviewed very thoroughly. The details are included in the uploaded file. For some reason, the first file I uploaded is missing page 8, but the discussion on this page needs also to be revised carefully.

Author Response
meat quality traits.
meat quality is very broad term with many traits, please specify
Thank you for your suggestion, I have revised it.
with meat quality traits
please specify the traits
Thank you for your suggestion, I have revised it.
genes
one or more gene are involved?
Thank you for your suggestion, I have revised it.
The expression of FOXO3 gene showed a decreasing trend in yak longissimus muscle at dif- ferent ages, with the highest expression at 6 months of age. E
there is no evidence for this anywhere in the manuscript
Thank you for your suggestion, I have revised it. After discussing with the subject members I realised that this part is not relevant to this study and have revised it.
please devide this sentence into two
Thank you for your suggestion, I have revised it.
limiting factor for the development
Thank you for your suggestion, I have revised it.
Page et al. 2002 (year) detected two SNPS in exon 14 and exon 9, respectively, 55 which were genotyped by the resource population and analyzed for genotype and shear 56 force values. Analysis of genotypes and shear force values in both populations revealed a 57 difference between paternal CAPN1 alleles in which the allele encoding isoleucine at po- 58 sition 530 and glycine at position 316 associated with decreased meat tenderness (in- 59 creased shear force values) relative to the allele encoding valine at position 530 and ala- 60 nine at position 316 (P < 0.05)[6]. Qiu et al. investigated the association of four SNPs in the insulin(INS) 61 gene with growth and developmental traits in chickens and found that one 62 SNP was highly significantly associated with small intestine length and another SNP was 63 also highly significantly associated with small intestine length and early daily weight 64 gain[7]. Jiang hui dentified 12 and 4 SNPs in male and female yaks, respectively, that could 65 be associated with body weight. 9 and 2 of these SNPs showed significant differences in yak body weight between genotypes at each locus in male and female yaks, respectively[8].
This is a very small summarization of the progress achieved during more than 20 years of SNP and QTL discovery. I suggest to generalize this more and refresh. Only reference in L64 and 65 are actually correlated to the manuscript theme
Thank you for your advice, I have rewritten and corrected that part.(Some content not relevant to this study has been modified, and some content relevant to this study has been added.)
2002 (year)
I think this should be numbered reference
Thank you for your advice, it was a mistake in the writing of the manuscript and I'm very sorry for the inconvenience caused to your reading. I have amended this section.
This reference should be cited according to journal propositions
Thank you for your suggestion, it has been modified.
please rephrase these sentenceses and cobine them in 1
Thank you for your suggestions, I have completed the changes.
82 KiM detected seven polymorphic sites 83 in the FOXO3 gene in 24 Koreans[12]. Wang Ling's study found that eight polymorphic loci 84 of the FOXO3 gene were present in each of the beef cattle populations Chen Biao et al. 85 used RNA-Seq to study the high and low tail pectoral muscle transcriptomes of recessive 86 white rock chickens (WRRh, WRRI) and Xinghua chickens (XHh, XHI) and found that a 87 total of 18 SNPs were identified in FOXO3[13]. In the mammalian ovary, the FOXO3 gene 88 can be associated with follicular atresia by promoting apoptosis in ovarian granulosa 89 cells[ The above findings suggest that the FOXO3 gene is rich in polymorphic infor- 14] mation.
please rephrase this paragraph to facilitate your hypothesis
Thank you for your suggestions, the changes have been made.
Etc
please add references here to confirm this sentence
Thanks for the suggestion, the reference has been added.
91 Improving meat quality and enhancing its palatability and edibility is of positive sig- 92 nificance to the development of yak industry. However, at present, the research on 93 FOXO3 gene mainly focuses on mice and human beings, and there is still a gap in the 94 study of the function of this locus as well as the relationship between the genetic variations 95 and the carcass traits of yaks. Therefore, the aim of this study was to analyse the genetic 96 effect of FOXO3 gene on yak carcass traits, and to use FOXO3 gene as an entry point to 97 randomly select some yak populations for genetic variation screening. And to explore this variation related to yak carcass and meat quality traits.
the same comment as above
Thank you for your suggestions, the changes have been made.
102 and the results are shown in Figure 1*: g.636C>G (SNP1), g.660T>C (SNP2), g.1296A>G (SNP3), g.1413C>G (SNP4) and g.1699C>G (SNP5).
please split this into 2 sentences
Thank you for your suggestions, the changes have been made.
115 The results of KASP typing at the FOXO3 gene SNPs locus are shown in Figure 2. The 116 results shown that three genotypes exist for all SNP1 ~ SNP5, namely the CC, GC, CC 117 genotype for SNP1, the CC, TC, TT genotype for SNP2, the AA, AG, GG genotype for 118 SNP3, the CC, CG, GG genotype for SNP4, and the CC, CG, GG genotype for SNP5, GG genotypes.
all figures and tables have to be self-explanatory. Unfortunately I do not see this. Please provide legend for pictures. Also provide information on number of individuals in every cluster you obtained.
Thanks for the suggestion, the graphic notes have been added.
please be more specific with the traits because only few of them were measured
CLR /% WLR /% REA /cm2
there are no explanation of abbreviations on the bottom of the table. Also please provide superscript letters in order to differences be visible. Please put p in the last column because in this way it cannot be read properly
Thanks for the suggestion, the feature about the measurement section has been added.
65.95±0.40 21.31±0.39 31.31±0.59 CG 57 5.48±0.20 66.07±0.81 19.69±0.78
please note that these are extremely high numbers of cooking loss and especially drip loss. Since I cannot find Liu et al., according to which this data are obtained, I don’t know how is the rate calculated nor are the figures right. But from logical point of view cooking loss of more than 65% is really really much. And it should not happen, especially in species close to bovine. As for the SNPs and their asssociation, one should bare in mind that tenderness is a trait that is very dependent on the age of the animal, so you should compare only genotypes within similar age. Since here age is not given, I do not know wheteher this was taken into account or not.
Thank you for your suggestion. The samples we collected all originated from Gannan, Gansu Province, where traditional grazing is practised and the cattle slaughtered are of adult age.
H2H2 and H2H3 had significantly lower shear forces than haplotype combination H1H1 and H1H2 (P<0.0
this should also be performed according to the age of the animals if the age was not the same for all of them
Thank you for your suggestion. The samples we collected all originated from Gannan, Gansu Province, where traditional grazing is practised and the cattle slaughtered are of adult age.
There was a significant difference in the expression in the longest dorsal muscle of 6-month-old yaks, 30-month- 179 old yaks, and 54-month-old yaks, with the highest expression of the FOXO3 gene at 6 180 months of age, followed by 30 months of age, and the lowest at 54 months of age. 181
so now it is obvioud that animals were not the same age. The difference is very big between them in terms of carcass and meat quality and this should be taken into accountz
Thank you for your suggestions, changes have been made.
The main indicators of meat quality are: pH, water retention,
Thank you for your suggestions, changes have been made.
Here the part of the manuscript is missing; I managed to download it again and read a page. The whole discussion part should be rewritten taking into account the results of this study and results
Thank you for your advice, which has helped me immensely with the manuscript, and I have completed the revisions as you requested. Please consult it this time, and if there are any improvements to be made, please suggest them, and I will continue to revise it. Thank you.
he genetic effects of haplo- type combinations of the FOXO3 gene on intramuscular fat content, muscle fibre diameter and muscle fibre density reached significant levels.
please rephrase this sentence
Thank you for your suggestions, I have completed the changes.
Studies have shown that the FOXO3 gene is widely expressed in chickens and yaks,
which studies?
Thank you for your suggestions, I have completed the changes.
it has been shown that the FOXO3 gene is a candi- date quantitative trait locus for muscle atrophy and lipid metabolism, which may affect meat quality traits in livestock and poultry[
this is a new sentence
Thank you for your suggestions, I have completed the changes.
316 drip loss rate (DLR; %), And cooking loss rate (CLR; %). DLR was measured using a mod- 317 ified filter paper press method described by Liu et al. WBSF and CLR were determined according to Shackelford et al. and Honikel, respectively.
This implies that there were some consecutive measurements, which I cannot see here? Also, please be careful about citations!
Thank you for your suggestion, if it wasn't for your suggestion I wouldn't have realised my error in the use of technical terms, in this part we measured the water loss rate, not the drip loss rate. Also regarding the cooking loss rate, we actually refer to the cooked meat rate in this part of the data.
Liu et al.
I cannot find this reference in the references
Thank you for the suggestion of the missing references in this section, I have added them.
pectrophotometer
please add manufacturer and country here
Thank you for your suggestion, I have revised it.
323 required concentration and purity were selected for mutation detec- tion
please add here or in the results how many samples passed quality control
Thank you for your suggestion, this part of the samples we have tested are all suitable before proceeding to the next test.
LTD.
please add country of origin
Thank you for your suggestions, I have completed the changes.
30S annealing
please add temperature here
Thanks for the suggestion, I have added the annealing temperature.
tissue
please specify tissues here
Thank you for your suggestion, I have revised it.
Trizol
please add the manufacturer and place of origin
Thank you for your suggestion, I have revised it.
spectrophotometer
which one?
Thank you for your suggestion, I have revised it.
agarose gel electrophoresis
what percentage?
Thank you for your suggestion, which I have explained in the manuscript.
355 Prime Script TM RT kit and gDNA reagent.
please add manufat+cturor and country of origin
Thank you for your suggestion, I have revised it.
376 Five polymorphisms in the FOXO3 gene were found in the yak. The association anal- 377 ysis of the polymorphisms (g.636C>G, g.1296A>G) revealed significant effects on WLR 378 and WBSF. Significant effects of haplotypes H2H3 on WBSF of yak meat quality. This re- 379 port will provide evidence that FOXO3 gene can be used as molecular markers for meat 380 quality traits in yak to improve meat quality traits in cattle through further marker-as- sisted selection.
please do not summarize as this is a conclusion section. Give the indications and conclusions
Thank you for your suggestion, I have revised it.
Round 2
Reviewer 1 Report
Comments and Suggestions for Authors
The revised version of the manuscript is tough to read due to the inclusion of all the editing (including the style or font modification).
Most of the remarks of the first review have NOT been carefully addressed.
Author Response
- The report by Youpeng Qi and colleagues describes the characterization of FoxO3 genotypes in 572 domestic Yaks (Bos grunniens) and the expression of the same gene in different tissues.
- The analysis of five SNPs was performed, and their related haplotypes were associated with meat technological traits.
- Although some results are interesting, the general structure of the work looks confusing, the link between the SNPs and the FOXO3expression analysis is unclear, and the biological interpretation of the findings is incomplete.
Thank you for your suggestions.SNPs and FOXO3 gene expression correlations and biological interpretations have been added in the manuscript
- Indeed, four of the five SNPs detected in exon II of FOXO3 (named 1 to 4 in the manuscript), are silent mutations that don’t result in any change in the protein structure. (By the way, the authors should recognize that SNPs 3 and 4 were already described by Wang, 2020, cited in references)
Crrected,Thank you for your suggestions, we agree with you. It has been enumerated in the text.
- The only explanation of a possible influence of SNPs 1-4 on FOXO3 pathway would be a change in the expression level of the transcript, due to a hypothetic destabilization of the mRNA or a codon-usage influence.
Crrected, Thank you for your suggestions, we agree with you.
- To demonstrate this, the expression level of the gene should be analyzed in a target tissue (e.g. longissimus dorsi) of a cohort of individuals carrying the different genotypes.
Thank you very much for your suggestion. It was not possible to carry out individual tissue expression tests for the relevant genotypes because the corresponding tissue expression samples were not collected from all tested yak individuals. We will certainly pay attention to this issue in the future.
- Conversely, the detection of the (rare) SNP 5 is worth consideration. Indeed, it looks responsible for one of the few FOXO3 aminoacidic variation between Bos taurus and Bos grunniens (the rare SNP5 “T” genotype reverts the protein sequence to that of cattle).This mutation by itself could be an interesting target for assisted selection, even though its rareness suggests a poor fitness for the harsh environment where yaks live.
Thank you for your suggestions, which have been revised in the manuscript.
- Minor remarks
Fig.1 should be organized in a single panel of better quality and the headings of the five chromatograms should be corrected (at present they look wrong)
Thank you for your suggestions, which have been revised in the manuscript.
At last,thank you for your review and suggestions on our manuscript. Those comments are all valuable and very helpful for revising and improving our paper. We have studied comments carefully and have made correction which we hope meet with approval.
Reviewer 2 Report
Comments and Suggestions for Authors
The manuscript underwent many improvements, taking into account the comments of the other reviewers. At the same time, only the summary contains many professional deficiencies and typing errors. In my opinion, they are:
line 2: „their“ instead of „its“
line 12: space is missing between them „force(WBSF)“
line 12: „Water“ with lowercase initial
line 13: space is missing between them “rate(WLR)”
line 15: space is missing between them “WBSF,WLR”
line 15-19: I recommend separating the two sentences.
line 17: The wording of "various yak tissues" is vague, especially as the Material and method chapter does not mention them, only the sample taken from the back muscle. This definitely needs to be replaced! At the same time, the purpose of blood sampling is not clear. The use of the term "genomic DNA" here can be considered an exaggeration (Genomic DNA constitutes the total genetic information of an organism), I think the authors only carry out FOXO3 genotyping from the individual biological samples (line 467).
line 19-20: It is not clear, how the expression of the FOXO3 gene changes depending on what!
line 19: I recommend writing the full Latin name of the long back muscle, musculus longissimus dorsi, with small initial letters.
line 24: "mutations" is more accurate instead of "variants".
There are still many typing errors in the manuscript, and it is the responsibility of the authors to correct them.
The objective must also be clearly stated, i.e. carcass traits are not the same as meat properties. They contain completely different valuable traits! Authors should make this clear (line 150-152).
Neither the objective nor the material and method reveal that the expression and haplo-, geno- and diplotypes of the FOXO3 gene will be examined at different ages.
I recommend a logical, all-encompassing wording of the Material and method chapter. The units of measurement must be written with a small initial letter, e.g. cm2, kg.
line 176: "SNP site" can be suggested instead of "SNP".
line 181: “population genetic polymorphism”, this term requires explanation.
line 183: “dominant genotypes”, this term requires explanation. According to our knowledge, the allele can be dominant. How do the authors know which allele is dominant and which is recessive, and what inter- and intralocal interactions there are? How can a heterozygous genotype be dominant? There are serious conceptual confusions in the manuscript! The word "dominant" can be misunderstood, the authors use the word "most frequent" instead.
Figure 3 appears twice in the manuscript. In Figure 3, the authors should replace "site identifier" with "SNP identifier".
For me, it is a huge methodological error that the meat quality parameters are not presented according to age, but only according to genotype. By the way, the title of Figure 5, in which the authors present expression by age (in which tissue?), is missing. Meat quality characteristics vary significantly with age. The meat quality characteristics of 6-month-old and 54-month-old animals should not be given with a single value. I miss the correction of phenotypic values for a given age. If the phenotypic value is not reliable, then it is doubtful to look for differences according to the genetic determination!
Author Response
The manuscript underwent many improvements, taking into account the comments of the other reviewers. At the same time, only the summary contains many professional deficiencies and typing errors. In my opinion, they are:
1.line 2: „their“ instead of „its“
Thank you for your suggestions, I have made the changes in the text.
line 12: space is missing between them „force(WBSF)“
Thank you for your suggestion, I have checked for errors and completed the changes.
line 12: „Water“ with lowercase initial
Thank you for your suggestion, I have revised it.
line 13: space is missing between them “rate(WLR)”
Thank you for your suggestion, I have revised it.
line 15: space is missing between them “WBSF,WLR”
Thank you for your suggestion, I have revised it.
line 15-19: I recommend separating the two sentences.
Thank you for your suggestions, I have completed the changes.
line 17: The wording of "various yak tissues" is vague, especially as the Material and method chapter does not mention them, only the sample taken from the back muscle. This definitely needs to be replaced! At the same time, the purpose of blood sampling is not clear. The use of the term "genomic DNA" here can be considered an exaggeration (Genomic DNA constitutes the total genetic information of an organism), I think the authors only carry out FOXO3 genotyping from the individual biological samples (line 467).
Thank you for your suggestions, I have completed the changes. The phrase "Various yak tissues" has been changed to "yak tissues", and the process of collecting yak tissue samples has been added to the Materials and Methods. The purpose of blood collection is to extract blood genomic DNA.
line 19-20: It is not clear, how the expression of the FOXO3 gene changes depending on what!
Thanks for the suggestion, I've corrected that part.
line 19: I recommend writing the full Latin name of the long back muscle, musculus longissimus dorsi, with small initial letters.
Thank you for your suggestions, I have completed the changes.
line 24: "mutations" is more accurate instead of "variants".
Thanks to your suggestion, I have checked the whole manuscript and replaced "Variants" with "mutations".
There are still many typing errors in the manuscript, and it is the responsibility of the authors to correct them.
The objective must also be clearly stated, i.e. carcass traits are not the same as meat properties. They contain completely different valuable traits! Authors should make this clear (line 150-152).
Thank you for the suggestion that this study did not analyse for carcass traits, that part was a writing error. We only analysed for the FOXO3 gene and meat quality traits.
Neither the objective nor the material and method reveal that the expression and haplo-, geno- and diplotypes of the FOXO3 gene will be examined at different ages.
I recommend a logical, all-encompassing wording of the Material and method chapter. The units of measurement must be written with a small initial letter, e.g. cm2, kg.
Thank you for your suggestions, I have completed the changes.
line 176: "SNP site" can be suggested instead of "SNP".
Thank you for your suggestions,I have completed the changes.
line 181: “population genetic polymorphism”, this term requires explanation.
Thanks for the suggestion, I've changed it to "genetic polymorphism".
line 183: “dominant genotypes”, this term requires explanation. According to our knowledge, the allele can be dominant. How do the authors know which allele is dominant and which is recessive, and what inter- and intralocal interactions there are? How can a heterozygous genotype be dominant? There are serious conceptual confusions in the manuscript! The word "dominant" can be misunderstood, the authors use the word "most frequent" instead.
Dear reviewer thanks for your suggestion, I have changed the term "dominant genotype" to "most frequent". I will pay attention to this part in the future, thank you.
Figure 3 appears twice in the manuscript. In Figure 3, the authors should replace "site identifier" with "SNP identifier".
Thank you for your suggestion this part has been modified and the modification has been completed in the diagram as well.
For me, it is a huge methodological error that the meat quality parameters are not presented according to age, but only according to genotype. By the way, the title of Figure 5, in which the authors present expression by age (in which tissue?), is missing. Meat quality characteristics vary significantly with age. The meat quality characteristics of 6-month-old and 54-month-old animals should not be given with a single value. I miss the correction of phenotypic values for a given age. If the phenotypic value is not reliable, then it is doubtful to look for differences according to the genetic determination!
Dear reviewer, Our yak samples were collected from a single region with the same level of feeding (all purely grazing yaks). All yaks were adults (young yaks are not slaughtered in the Tibetan Plateau region), so we analysed age as a fixed factor. Thank you for your suggestion, and we will take your advice and try to do the correlation analysis of meat quality traits among yaks of different ages in our subsequent work. As a side note, Figure 5 shows that we subsequently collected tissue samples from yaks of three ages to look at the expression of the FOXO3 gene in the muscles of yaks of different ages, which I found to be irrelevant to the content of this study, and has been deleted as a correction to Figure 5.
Reviewer 3 Report
Comments and Suggestions for Authors
The article quality increased tremendously after the previous review. I have only a couple of suggestions that need to be corrected:
L151: Table 6 further explores how the presence/absence of specific haplotypes was linked to meat traits. In this case, the presence of the H2 haplotype was associated with increased WBSF (P=0.009), - but when the haplotype is absent the WBSF is higher, not lower. Also please note that lower WBSF means increased tenderness, and not the other way around.
L187: The reference J H LEE should be corrected according to the journal propositions
L196: Mette Sorensen? This is also a reference? If yes, please cite according to the journal rules
L297 At 48 h postmortem, the rib eye area (REA; cm2) was traced from the longissimus dorsi – please change traced to determined
L301: water loss rate (WLR; %) is actually water holding capacity (according to description of the Liu et al.-the reference that you provided) please change WLR to WHC (water holding capacity) throughout the manuscript. Additionally, if the method you used is a modification of the method by Liu et al. Please describe in brief the modification.
Author Response
Manuscript Number: ijms-2759716
Article Title: Mutations in the FOXO3 gene and their effects on meat traits in Gannan yaks
Journal Title: International Journal of Molecular Sciences
To Reviewers and Editor:
We sincerely thank the editor and all reviewers for their valuable feedback on improving the quality of our manuscript. We have made extensive corrections to our previous draft which we hope meet with approval. The reviewer comments are laid out below in italicized font and specific concerns have been numbered. Our response is given in red font and changes/additions to the manuscript are given in the red text. The detailed corrections are listed below.
Q1: The article quality increased tremendously after the previous review. I have only a couple of suggestions that need to be corrected:
L151: Table 6 further explores how the presence/absence of specific haplotypes was linked to meat traits. In this case, the presence of the H2 haplotype was associated with increased WBSF (P=0.009), - but when the haplotype is absent the WBSF is higher, not lower. Also please note that lower WBSF means increased tenderness, and not the other way around.
Thank you for your suggestions, I have completed the changes.[L157]
Q2: L187: The reference J H LEE should be corrected according to the journal propositions.
Thank you for your suggestions, I have completed the changes. And removed this document already.
Q3: L196: Mette Sorensen? This is also a reference? If yes, please cite according to the journal rules.
Thank you for your suggestions, I have completed the changes.[L180]
Q4: L297 At 48 h postmortem, the rib eye area (REA; cm2) was traced from the longissimus dorsi -please change traced to determined
Thank you for your suggestions I have made the changes.[L281]
Q5: L301: water loss rate (WLR; %) is actually water holding capacity (according to description of the Liu et al.-the reference that you provided) please change WLR to WHC (water holding capacity) throughout the manuscript. Additionally, if the method you used is a modification of the method by Liu et al. Please describe in brief the modification.
Thank you for your suggestion. I have completed the modifications. Regarding the cooked meat rate, I actually modified it to reflect cooking losses.
Round 3
Reviewer 1 Report
Comments and Suggestions for Authors
The manuscript looks formally improved from the previous version, but some part, in particular the discussion, are still confusing and inaccurate.
Some statements are contradictory with regard to the cited references.
For Example, at raws 249-251, it seems that FOXO3 upregulation is a positive factor for meat production but, according to the cited literature, it is involved in muscle atrophy.
The discussion must be shortened and critically reviewed, avoiding repetitive assertions and the description of the state of the art (already made in the introduction). Instead, the authors should highlight the novelty and the importance of their results and identify the limits of the study.
minor remarks:
raw 173 and 255: why "EELEN " is uppercase?
Raw 187: why J H LEE is uppercase?
Author Response
Manuscript Number: ijms-2759716
Article Title: Mutations in the FOXO3 gene and their effects on meat traits in Gannan yaks
Journal Title: International Journal of Molecular Sciences
To Reviewers and Editor:
We sincerely thank the editor and all reviewers for their valuable feedback on improving the quality of our manuscript. We have made extensive corrections to our previous draft which we hope meet with approval. The reviewer comments are laid out below in italicized font and specific concerns have been numbered. Our response is given in red font and changes/additions to the manuscript are given in the red text. The detailed corrections are listed below.
Q1: The manuscript looks formally improved from the previous version, but some part, in particular the discussion, are still confusing and inaccurate.
Some statements are contradictory with regard to the cited references.
For Example, at raws 249-251, it seems that FOXO3 upregulation is a positive factor for meat production but, according to the cited literature, it is involved in muscle atrophy..
Thank you for your suggestion. I have completed the modifications regarding this part in the text.
Q2: The discussion must be shortened and critically reviewed, avoiding repetitive assertions and the description of the state of the art (already made in the introduction). Instead, the authors should highlight the novelty and the importance of their results and identify the limits of the study.
Thank you for your suggestion. I have read the entire text and completed the revisions. The repetitive content in the introduction has also been revised, and the focus on the innovation of the research and future work has been proposed. [L259-L262]
Q3: minor remarks:
raw 173 and 255: why "EELEN " is uppercase?
Thank you for your suggestion. I have completed the modifications.[L230]
Q4: Raw 187: why J H LEE is uppercase?
Thank you for your suggestion. I have completed the modifications.
Reviewer 2 Report
Comments and Suggestions for Authors
The authors still do not write in the manuscript what age the examined animals were. They do not specify the age at which the meat quality traits, which are evaluated depending on the genotype, were corrected. I do not accept the authors' answer. The second version (previous) of the manuscript had Figure 5, which has now been deleted. This presented the expression results of individuals aged 6 - 54 months. Where did these go? The authors replied that the animals were the same age after all. If this is the case, why is it necessary to take into account the effect of age in the model (GLM) they used?
Similarly, the manuscript does not reveal at what age the expression studies of the FOXO3 gene were performed in the tissue samples taken.
Both depend on age, so this information is essential. Age is not mentioned in any cited source work as well.
The third version (current) of the manuscript does not show the corrections, which makes it difficult for the reviewer to check. In addition to all this, the manuscript still needs correction of typing errors in countless places.
Until these are precisely defined and corrected, I cannot accept the manuscript for publication.
Author Response
Manuscript Number: ijms-2759716
Article Title: Mutations in the FOXO3 gene and their effects on meat traits in Gannan yaks
Journal Title: International Journal of Molecular Sciences
To Reviewers and Editor:
We sincerely thank the editor and all reviewers for their valuable feedback on improving the quality of our manuscript. We have made extensive corrections to our previous draft which we hope meet with approval. The reviewer comments are laid out below in italicized font and specific concerns have been numbered. Our response is given in red font and changes/additions to the manuscript are given in the red text. The detailed corrections are listed below.
Q1: The authors still do not write in the manuscript what age the examined animals were. They do not specify the age at which the meat quality traits, which are evaluated depending on the genotype, were corrected. I do not accept the authors' answer. The second version (previous) of the manuscript had Figure 5, which has now been deleted. This presented the expression results of individuals aged 6 - 54 months. Where did these go? The authors replied that the animals were the same age after all. If this is the case, why is it necessary to take into account the effect of age in the model (GLM) they used?
Thank you for your suggestion. First of all, I apologize for the wording issues and chart deletion issues that have caused difficulties in reviewing my paper.The yaks used for KASP detection in this study were all from adult yaks aged 5-6 in Gannan Tibetan Autonomous Prefecture, with 523 yaks aged 6 and 49 yaks aged 5.So we uniformly analysed them as adult cattle without the effect of differences between ages.[L270]
Regarding Fig. 5 Expression of FOXO3 gene in muscle at different ages I have deleted it, the reason for deletion is as follows: it has been reported that FOXO3 gene is associated with muscle atrophy in animals, in view of this we have done the analysis of the expression of the longissimus dorsi muscle of yaks aged from 6-54 months, and our findings show that the expression of FOXO3 gene is down-regulated with age, and that the change in the amount of expression does not explain the characteristics of muscle atrophy . Therefore, Figure 5 was deleted.
Thanks for the suggestion, I have removed the age effect from the model.
Q2: Similarly, the manuscript does not reveal at what age the expression studies of the FOXO3 gene were performed in the tissue samples taken.
Thanks to your suggestion, I have indicated the age of the tissue sample donor yak in the manuscript.
Q3: Both depend on age, so this information is essential. Age is not mentioned in any cited source work as well.
Thank you for your suggestion about the age of the yak corresponding to the tissue expression samples I have modified it in the material methods. [L276-L279]
Q4: The third version (current) of the manuscript does not show the corrections, which makes it difficult for the reviewer to check. In addition to all this, the manuscript still needs correction of typing errors in countless places.
Thank you for your suggestions I have completed the changes.
Round 4
Reviewer 2 Report
Comments and Suggestions for Authors
The manuscript has improved overall, the previous doubts have been reduced for future readers. I still recommend correcting typographical errors in the manuscript before publication. Furthermore, it should be emphasized that the processing was carried out under the usual fattening conditions (including the age at the end of fattening) of the yak.
Author Response
Manuscript Number: ijms-2759716
Article Title: Mutations in the FOXO3 gene and their effects on meat traits in Gannan yaks
Journal Title: International Journal of Molecular Sciences
To Reviewers and Editor:
We sincerely thank the editor and all reviewers for their valuable feedback on improving the quality of our manuscript. We have made extensive corrections to our previous draft which we hope meet with approval. The reviewer comments are laid out below in italicized font and specific concerns have been numbered. Our response is given in red font and changes/additions to the manuscript are given in the red text. The detailed corrections are listed below.
Q1: The manuscript has improved overall, the previous doubts have been reduced for future readers. I still recommend correcting typographical errors in the manuscript before publication.
Thank you for your suggestion. The typographical errors in the manuscript have been corrected and marked in red.
Q2: Furthermore, it should be emphasized that the processing was carried out under the usual fattening conditions (including the age at the end of fattening) of the yak.
Thank you for your suggestion.
The Qinghai Tibet Plateau has high altitude and a shortage of cold season forage. Yaks are generally not fed fodder and are reared by grazing and are usually slaughtered when they are 5-6 years old. We have completed the modifications.[L268-L270]